# Divergent functions of two clades of flavodoxin in diatoms mitigate oxidative stress and iron limitation

**Shiri Graff van Creveld[1], Sacha N Coesel[1], Stephen Blaskowski[1,2], Ryan D Groussman[1], Megan J Schatz[1], E Virginia Armbrust[1]\***

[1]School of Oceanography, University of Washington, Seattle, United States;
[2]Molecular Engineering Graduate Program, University of Washington, Seattle, WA, Seattle, United States

**Abstract** Phytoplankton rely on diverse mechanisms to adapt to the decreased iron bioavailability and oxidative stress-inducing conditions of today's oxygenated oceans, including replacement of the iron-requiring ferredoxin electron shuttle protein with a less-efficient iron-free flavodoxin under iron-limiting conditions. Yet, diatoms transcribe flavodoxins in high-iron regions in contrast to other phytoplankton. Here, we show that the two clades of flavodoxins present within diatoms exhibit a functional divergence, with only clade II flavodoxins displaying the canonical role in acclimation to iron limitation. We created CRISPR/Cas9 knock-outs of the clade I flavodoxin from the model diatom *Thalassiosira pseudonana* and found that these cell lines are hypersensitive to oxidative stress, while maintaining a wild-type response to iron limitation. Within natural diatom communities, clade I flavodoxin transcript abundance is regulated over the diel cycle rather than in response to iron availability, whereas clade II transcript abundances increase either in iron-limiting regions or under artificially induced iron limitation. The observed functional specialization of two flavodoxin variants within diatoms reiterates two major stressors associated with contemporary oceans and illustrates diatom strategies to flourish in diverse aquatic ecosystems.

**\*For correspondence:**
armbrust@uw.edu

**Competing interest:** The authors declare that no competing interests exist.

## Editor's evaluation

This study presents valuable findings, with solid evidence, regarding the functional diversification of flavodoxins from diatoms, a protein initially described as an Fe-sparing substitute for ferredoxin in Fe-poor open ocean environments.

## Introduction

All of life depends on redox-based metabolic pathways driven by the transfer of electrons between protein donors and acceptors. One of the abundant electron shuttles in phototrophs is ferredoxin (Fd), a small, soluble iron–sulfur-cluster-containing protein that accepts electrons from the stromal surface of photosystem I during oxygenic photosynthesis and facilitates transfer to acceptors involved in diverse metabolic processes (*Mondal and Bruce, 2018*). Proteins such as Fds with iron–sulfur clusters are ancient biocatalysts hypothesized to have arisen when oxygen was largely absent, and ferrous iron and sulfide were plentiful (*Cammack, 1982*). During the great oxidation event approximately 2.4 billion years ago, ferrous iron became oxidized to the ferric form and rapidly precipitated either as ferric hydroxide or as insoluble complexes with anionic salts. Microbial growth in aerobic habitats thus became limited by iron bioavailability (*Imlay, 2006*), as found in about a third of today's oceans (*Behrenfeld and Milligan, 2013*; *Boyd et al., 2007*; *Moore et al., 2013*). Moreover, iron-containing

proteins are sensitive to damage via oxygen and reactive oxygen species (ROS), and Fd is downregulated in response to oxidative stress (*Singh et al., 2010*; *Singh et al., 2004*). Yet, contemporary marine phytoplankton (cyanobacteria and photosynthetic eukaryotes) continue to rely on Fd and may sequester up to 30–40% of their cellular iron within these proteins (*Erdner, 1997*). Reliance on Fd for electron shuttling during oxygenic photosynthesis thus carries with it both a requirement for sufficient iron bioavailability and an enhanced sensitivity to the ROS generated during photosynthesis.

Under iron-limiting conditions, photosynthetic microbes commonly replace the iron-containing Fd with flavodoxin, a functional homologue (*Smillie, 1965*; *Hutber et al., 1977*; *Sandmann et al., 1990*). Flavodoxin uses a flavin mononucleotide (FMN) as the co-factor and is a less efficient electron shuttle, which nonetheless allows continued photosynthetic electron transport when iron is scarce (*Zumft and Spiller, 1971*; *Yoch and Valentine, 1972*). The absence of flavodoxin from land plants and certain coastal algal species led to the premise that flavodoxin is lost from species in iron-rich environments, including terrestrial systems (*Pierella Karlusich et al., 2015*). However, the downregulation of Fd in response to adverse conditions (*Singh et al., 2010*; *Singh et al., 2004*) and the induction of flavodoxin in the cyanobacteria *Synechocystis* under oxidative stress (*Jeanjean et al., 2003*; *Kojima et al., 2006*; *Singh et al., 2004*) suggests additional physiological roles for flavodoxin within phytoplankton.

Expression levels of flavodoxin relative to Fd (inferred either from transcript or protein levels) have been used in several studies to evaluate whether natural communities of marine phytoplankton experience iron limitation in situ (*Boyd et al., 1999*; *DiTullio et al., 2005*; *Erdner and Anderson, 1999*; *Jones, 1988*). Metatranscriptome studies indicate that, as expected, most major eukaryotic phytoplankton lineages (chlorophytes, haptophytes, and dinoflagellates) regulate flavodoxin transcript abundances in response to iron availability, with reduced flavodoxin transcript levels in iron-replete environments (*Caputi et al., 2019*; *Carradec et al., 2018*). A closer examination of environmental metatranscriptomes (*Marchetti et al., 2012*; *Carradec et al., 2018*; *Caputi et al., 2019*) finds that diatom flavodoxin transcript abundances are not inversely correlated with iron availability as expected, and flavodoxins are often detected at high abundances under iron-replete conditions. Although direct comparisons between environmental flavodoxin and Fd transcript abundances are complicated by the fact that Fd is encoded on the plastid genomes of some diatoms (*Groussman et al., 2015*; *Gueneau et al., 1998*; *Lommer et al., 2010*; *Oudot-Le Secq et al., 2007*; *Roy et al., 2020*), diatoms appear to regulate the transcription of flavodoxin differently compared to other phytoplankton. A potential explanation for the distinctive flavodoxin transcriptional patterns of natural communities of diatoms comes from the discovery in a few model diatoms that flavodoxin diverged into two distinct phylogenetic clades with only those flavodoxins from clade II induced upon iron limitation, suggesting that clade I flavodoxins may play a different role within these species (*Whitney et al., 2011*).

Here, we combine genetic surveys of environmental and experimental datasets with gene knock-out strategies in a model diatom to elucidate the differentiation of clade I and clade II flavodoxins in phytoplankton and their potential roles in diatoms. Our results from both model diatoms and natural phytoplankton communities indicate that these two flavodoxin clades represent a functional divergence where the clade II flavodoxins in stramenopiles play a role in acclimation to low-iron conditions and the clade I variant likely conveys acclimation to oxidative stresses. This functional divergence likely augments the ability of diatoms to flourish in diverse ecosystems.

## Results

### Clade I flavodoxins emerged within a subset of stramenopiles

The original description of clade I and II flavodoxins was based on sequences from six model diatoms and one non-diatom stramenopile (*Whitney et al., 2011*), with the potential for functional distinction derived from transcriptional responses of two *Thalassiosira* diatoms, each encoding a different clade of flavodoxin. We supplemented this work by examining publicly available transcript data and found that in the three model diatoms that encode both clade I and clade II flavodoxins, only the clade II flavodoxins were significantly induced upon iron limitation (*Graff van Creveld et al., 2016*; *Lommer et al., 2012*; *Mock et al., 2017*; *Smith et al., 2016*; summed in *Supplementary file 1a*), reiterating the potential for different functional roles of these proteins.

To determine the distribution of clade I and clade II flavodoxins in additional marine phytoplankton, we screened publicly available genomes and transcriptomes of over 500 marine protists, bacteria, and

archaea (*Coesel et al., 2021*) for flavodoxin sequences using a custom-made hidden Markov model (hmm)-profile of the flavodoxin domain adapted from PF00258 (e < 0.001; hmmsearch; *Eddy, 1998*; *Finn et al., 2014*). The flavodoxin domain was detected in 1191 flavodoxin genes (*Supplementary file 2*; supplementary fasta file 2) and included 332 genes that clustered with known photosynthetic flavodoxin proteins (*Figure 1—figure supplement 1A*, black labels); the remaining sequences displayed similarity to a diverse and distant clade of non-photosynthesis-related proteins (*Figure 1—figure supplement 1A*, red labels). We further used this phylogenetic tree and the distinction between the known photosynthetic flavodoxins (*Figure 1—figure supplement 1A*, black labels, the right side of the tree) to the other genes (*Figure 1—figure supplement 1A*, red labels, the left side of the tree) to identify photosynthetic flavodoxins that were not previously characterized (unlabeled flavodoxins on the right side of the tree). A similar method was used in *Caputi et al., 2019*. The presumed photosynthesis-associated flavodoxins segregate into four clades (*Figure 1A*, *Figure 1—figure supplement 1A*). Two clades were composed of photosynthetic flavodoxins from green algae, one that grouped with gamma-proteobacteria (highlighted in pink, *Figure 1A*, *Figure 1—figure supplement 1B*) and another that grouped with cyanobacteria and dinoflagellates (highlighted in green, *Figure 1A*, *Figure 1—figure supplement 1B*). Within these clusters, the labeled *Synechococcus* and *Prochlorococcus* flavodoxins were previously shown to be induced in response to iron limitation (*Kashtan et al., 2014*; *Mackey et al., 2015*; *Thompson et al., 2011*; *Yousef et al., 2003*). One iron limitation-induced *Synechococcus* flavodoxin grouped within the clade of green algae and gamma-proteobacteria flavodoxins (highlighted in pink, *Figure 1A*, *Figure 1—figure supplement 1B*); all other iron limitation-induced *Synechococcus* flavodoxins clustered within the green algae clade that contained proteins from cyanobacteria and dinoflagellates (highlighted in green, *Figure 1A*, *Figure 1—figure supplement 1B*).

Clade I and II flavodoxins, as originally identified by *Whitney et al., 2011*, were detected exclusively in eukaryotic algae originally derived from a secondary endosymbiosis of a red alga (*Figure 1A*, *Figure 1—figure supplement 1B*). Clade I flavodoxins were detected in stramenopiles (highlighted in orange, *Figure 1*, *Figure 1—figure supplement 1B*) and two dinoflagellates (*Durinskia baltica* and *Kryptoperidinium foliaceum*). Both dinoflagellates express proteins originating from their diatom endosymbionts (*Hehenberger et al., 2016*; *Imanian et al., 2010*; *Imanian and Keeling, 2007*; *Yamada et al., 2019*). The flavodoxin from the stramenopile *Aureococcus anophagefferens*, previously defined as a clade I flavodoxin (*Whitney et al., 2011*), groups with another unclassified pelagophyte (*Figure 1A*, *Figure 1—figure supplement 1B*). This branch is defined here as the base of clade I. None of the previously tested clade I flavodoxin genes transcriptionally respond to iron limiting growth conditions (stars, *Figure 1A*, *Supplementary file 1a*). Clade II flavodoxins are phylogenetically more diverse than clade I flavodoxins and consist of sequences from diatoms, non-diatom stramenopiles, haptophytes, and dinoflagellates (*Karenia brevis* and *Karlodinium micrum*) (*Figure 1A*, *Figure 1—figure supplement 1B*). The base of clade II is defined here by a cluster of haptophyte sequences that includes a *Phaeocystis* flavodoxin (labeled) experimentally induced under iron limitation (*Wu et al., 2019*). Known diatom iron-limitation-responsive flavodoxins all cluster within clade II (*Supplementary file 1a*), consistent with a functional divergence between clade I and clade II flavodoxins within the stramenopiles.

To determine the distribution of clade I and clade II flavodoxins within the diverse stramenopile lineage more thoroughly, we queried 56 additional publicly available stramenopile transcriptomes and genomes (*Supplementary file 1b*, *Supplementary file 2*; supplementary fasta file 3). Flavodoxins from both clades appear to be similarly distributed amongst open ocean and coastal isolates, with no evidence that clade II flavodoxins are retained only in species isolated from low-iron environments (*Figure 1—figure supplement 1B*, *Supplementary file 1b*). Diatom clade I flavodoxins from pennate and centric diatoms form a tight cluster with high bootstrap support (*Figure 1B*, *Figure 1—figure supplement 1C*), with multiple copies detected in a subset of species (*Supplementary file 1b*, *Figure 1—figure supplement 1D*). Many examined taxa encode both clade I and clade II flavodoxins. About half of the taxa appear to encode a flavodoxin from only one clade (*Figure 1—figure supplement 1D*), although this absence may reflect culture conditions rather than absence from their respective genomes as most available sequence data are derived from transcriptomes (*Supplementary file 1b*). Nonetheless, the absence of clade I flavodoxin sequences appears more common in stramenopiles that are more distantly related to diatoms, such as *Phaeomonas* (Pinguiophyceae; *Figure 1—figure*

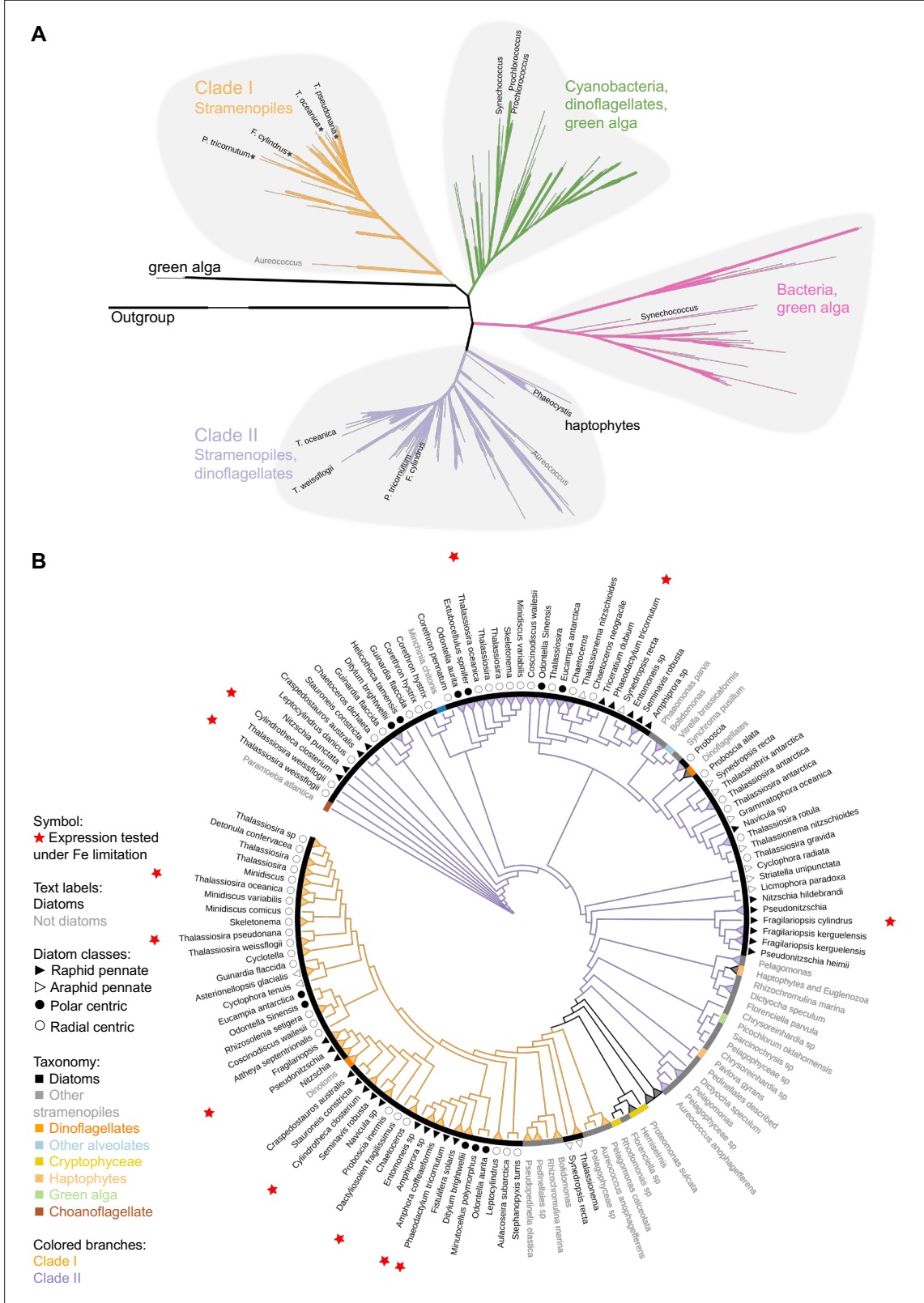

**Figure 1.** Phylogeny of flavodoxins in marine microorganisms. (**A**) Maximum-likelihood (RAxML) phylogenetic tree of photosynthetic flavodoxins (see *Figure 1—figure supplement 1A* for the full tree, *Figure 1—figure supplement 1B* for taxonomic annotation); non-photosynthetic flavodoxins as well as flavodoxin domains in other proteins are collapsed as an 'outgroup.' Thick branch lines represent bootstrap support greater than 0.7. Different clades are marked with colors: clade I (orange), clade II (purple), bacteria/green algae clade (pink), cyanobacteria/ dinoflagellate/green alga clade

*Figure 1 continued on next page*

*Figure 1 continued*

(green). Labeled taxa indicated with black labels were experimentally tested for response to iron limitation; tested flavodoxins indicated with black starts were not induced (***Supplementary files 1 and 2***). *Aureococcus* (indicated in gray) has not been experimentally tested in response to iron limitation. (**B**) Unrooted maximum-likelihood (RAxML) phylogenetic tree of clade I and II flavodoxins with 56 additional stramenopile strains (listed in ***Supplementary file 1b***). Thick branch lines represent bootstrap support greater than 0.7. Branches collapsed (indicated with triangles at tips) for non-stramenopiles and at genus level within stramenopiles (see ***Figure 1—figure supplement 1C*** for the full tree). Branch colors represent clade (I, II, orange and purple, respectively). Colored strip represents taxonomy, dinoflagellates with diatom endosymbionts are labeled 'dinotoms.' Red stars represent flavodoxins with expression experimentally tested under iron limitation.

The online version of this article includes the following figure supplement(s) for figure 1:

**Figure supplement 1.** Flavodoxins in publicly available databases.

---

***supplement 1E***), while stramenopiles more closely related to diatoms, such as members of Pelagophyceae and Dictyochophyceae, commonly encode both clade I and clade II flavodoxins (***Figure 1B***, ***Figure 1—figure supplement 1E***, ***Supplementary file 1b***).

To explore a potential molecular basis for the clade I and II protein divergence, we aligned stramenopile flavodoxins to the flavodoxin sequence of the red alga *Chondrus crispus* (***Supplementary file 2***; supplementary fasta file 4), as its structure is well studied and the FMN-binding sites were previously identified (***Fukuyama et al., 1992***). As expected, amino acid side chains that form hydrogen bonds with the FMN (underlined in ***Figure 1—figure supplement 1F***) are conserved between the two clades. The clade I and clade II flavodoxin sequences differ from each other at amino acids 57 and 103 (numbered according to *C. crispus* sequence, ***Figure 1—figure supplement 1F***). At amino acid 57, the dominant amino acid is asparagine (N) in *C. crispus* and clade I proteins, and histidine (H) in clade II proteins. At amino acid 103, the dominant amino acid is cysteine (C) in *C. crispus* and clade II proteins, and alanine (A) in clade I proteins. Both distinguishing amino acid changes occur near the FMN-binding tryptophan and tyrosine (W56, Y98, indicated with asterisks in ***Figure 1—figure supplement 1F***), consistent with distinct functions for the two flavodoxins.

## Diatom flavodoxins transcriptional responses to oxidative and iron stress

Given the observed dual role of the cyanobacterial flavodoxin in adaptation to iron limitation and oxidative stress (***Jeanjean et al., 2003***; ***Kojima et al., 2006***; ***Singh et al., 2004***), we hypothesized that

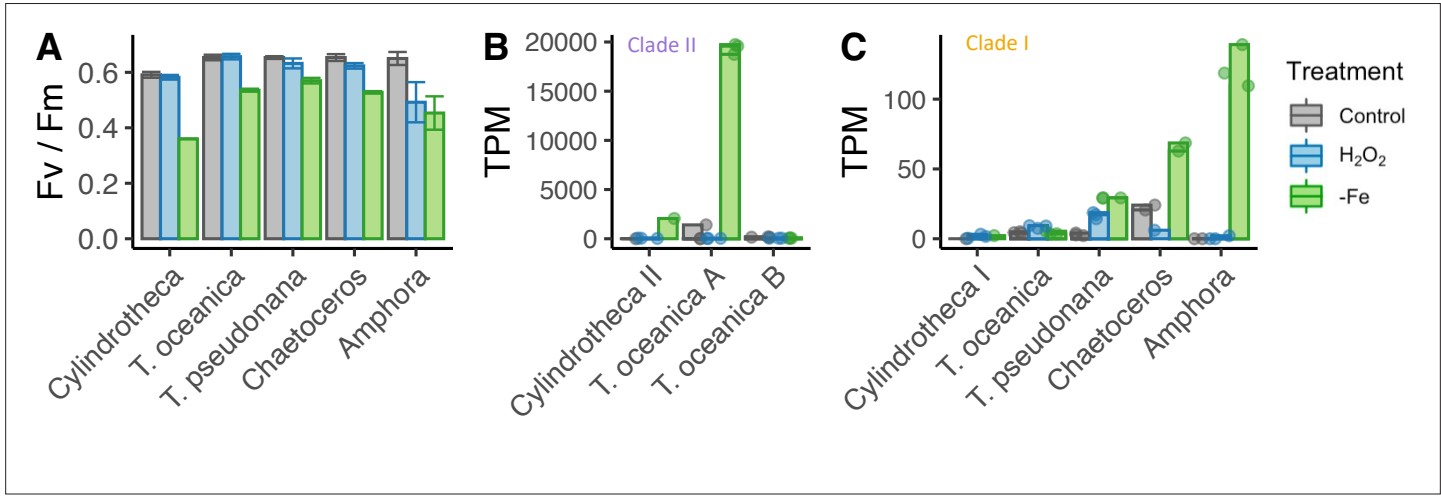

**Figure 2.** Iron limitation and oxidative stress in diatom cultures. (**A**) Photosynthetic efficiency (Fv/Fm) of five diatom species, before harvesting the cells for each transcriptome, error bars represent the standard deviation of biological triplicates. Transcripts per million (TPM) of clade II (**B**) and clade I (**C**) flavodoxins in response to iron limitation and $H_2O_2$ treatment. *T. oceanica* clade II flavodoxins, previously named FLDA1 and FLDA2 (***Whitney et al., 2011***), are marked here as A and B, respectively. *Cylindrotheca* encodes one flavodoxin from each clade, marked here I and II according to the clade. Note the different Y axis scales in panels (**B**) and (**C**). Individual measurements are marked in circles, maximal values in colored bars.

The online version of this article includes the following figure supplement(s) for figure 2:

**Figure supplement 1.** Diatoms cultures transcriptomes.

the two clades in stramenopiles represented a functional divergence with the iron limitation response specific to clade II, and the oxidative stress response specific to clade I flavodoxins. To test this hypothesis, we examined transcriptional responses of five diatom species exposed to either oxidative stress or to iron limitation. In order to facilitate screening of multiple diatoms for whether they transcribed clade I and II flavodoxins in response to iron limitation, we used the strong iron-chelator desferrioxamine B (DFB) and enhanced short-term iron limitation. We chose two closely related model centric diatoms: the estuarian-isolated *Thalassiosira pseudonana* and the open ocean-isolated *Thalassiosira oceanica*, as well as three non-model open ocean-isolated diatoms: two pennates, *Amphora coffeaeformis* and *Cylindrotheca closterium* and one centric *Chaetoceros* sp., none of which had publicly available genome or transcriptome sequences. Again, in order to facilitate screening of multiple diatoms, we focused on a single treatment for oxidative stress. Oxidative stress was induced by the lowest lethal dose of $H_2O_2$ (200–250 µM), as a similar treatment was shown to be representative of other, environmentally relevant form of oxidative stressors in *T. pseudonana* and *Phaeodactylum* (*Graff van Creveld et al., 2015*; *Mizrachi et al., 2019*; *Volpert et al., 2018*). For each diatom, six replicates were grown in iron-replete conditions and three replicates in iron-limiting conditions until the low-iron cultures displayed a decrease in maximum photochemical yield of photosystem II (Fv/Fm), 3–6 d (depending on species, *Figure 2—figure supplement 1A–C*, *Figure 2A*, *Supplementary file 1c*), indicative of iron limitation, at which point transcriptome samples were collected for both the iron-limited and iron-replete conditions. Three of the iron-replete replicates were exposed to oxidative stress, mimicked by a lethal dose of $H_2O_2$, and transcriptome samples were collected about 1.5 hr after exposure, when the cell phenotype (Fv/Fm or cell abundance) was unaltered from control. The three conditions elicited distinct overall transcriptional responses in the five diatom species as seen in multidimensional scaling (MDS) plots (*Figure 2—figure supplement 1D*).

The different diatom species expressed different flavodoxins clades. *Cylindrotheca* and *T. oceanica* transcribed both clade I and II flavodoxins; *T. oceanica* transcribed two isoforms of clade II flavodoxins (labeled here A, B, *Figure 2B*); *T. pseudonana*, *Chaetoceros*, and *Amphora* transcribed only clade I flavodoxins. The induction of flavodoxins in response to oxidative or iron stress also varied between the different species. The transcription levels of clade II genes from *T. oceanica* (clade II-A) and *Cylindrotheca* were significantly elevated in response to iron limitation (5.3- and 9.5-fold change, respectively, false discovery rate [FDR] <0.01, *Figure 2B*), whereas the clade I genes in these diatoms did

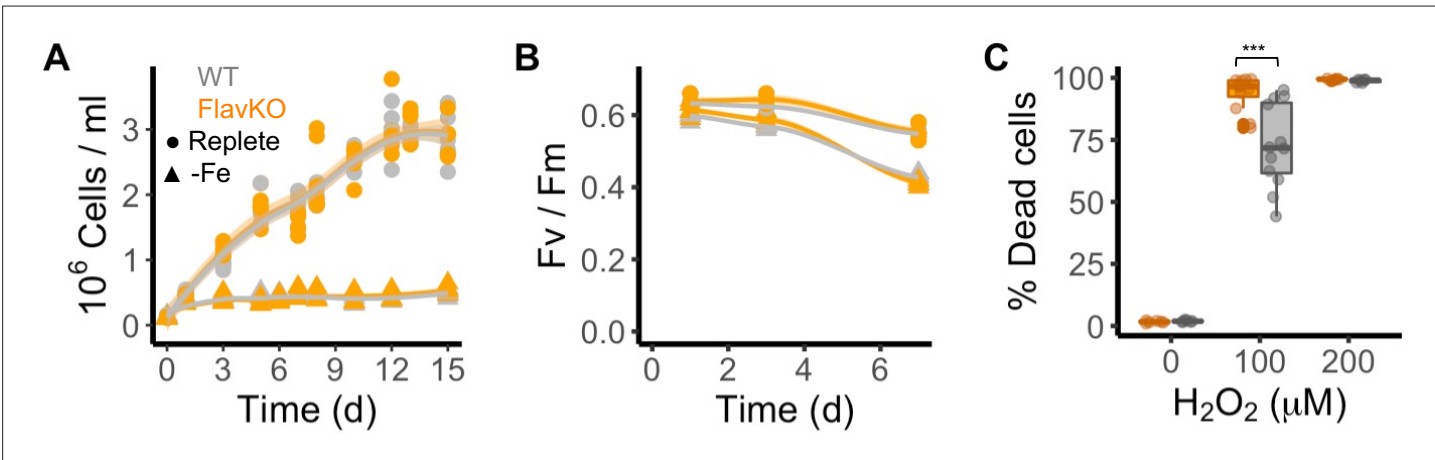

**Figure 3.** Response of *T. pseudonana* flavodoxin KO and WT lines to iron limitation and oxidative stress. Three independent WT (gray) and flavodoxin KO (orange) lines were grown under iron- replete (circles) and iron-limiting (triangles) conditions for several days. (**A**) Cell abundance, measured by flow cytometry. (**B**) Photosynthetic efficiency, measured by phytoPAM. Individual measurements marked in symbols, means of triplicates in lines. (**C**) Percentage of Sytox Green-positive (dead) cells, measured by flow cytometry 24 hr after treatment with 0, 100, or 200 µM $H_2O_2$. Box plots combine two independent experiments, ANOVA between WT and KO (indicated with ***), significant 9 • $10^{-5}$.

The online version of this article includes the following source data and figure supplement(s) for figure 3:

**Figure supplement 1.** Flavodoxin knock-outs in *T. pseudonana.*

**Figure supplement 1—source data 1.** Uncropped gel of *Figure 3—figure supplement 1B*.

**Figure supplement 1—source data 2.** Uncropped gel of *Figure 3—figure supplement 1B*.

not display a significant change in transcript abundance either after exposure to $H_2O_2$ or under iron limitation (FDR = 1, *Figure 2C*). The *T. oceanica* clade II-B gene transcript levels were not responsive to either treatment (*Figure 1B*), in agreement with previous results (*Supplementary file 1a*, *Lommer et al., 2012*). The transcribed clade I flavodoxins of *T. pseudonana*, *Chaetoceros,* and *Amphora* were elevated under iron limitation (3.2-, 1.5-, and 9.9-fold change, respectively, FDR < 0.01, *Figure 2C*). These diatoms encode only a clade I flavodoxin, suggesting that clade I flavodoxins are induced in iron limitation only when clade II flavodoxins are absent. Only the *T. pseudonana* clade I flavodoxin gene was significantly differentially transcribed in response to $H_2O_2$ treatment (2.4-fold increase, FDR < 0.01, *Figure 2C*). Together, these results confirm that clade II flavodoxins transcriptionally respond to iron limitation and suggest that the transcriptional response of the clade I flavodoxin within a species may depend on whether clade II flavodoxin is also transcribed.

## Functional role of a clade I flavodoxin

To disambiguate the potential for functional redundancy or synergy between clade I and II flavodoxins within a species, we generated a clade I flavodoxin gene knock-out (KO) in *T. pseudonana* as this diatom encodes a single copy of the clade I gene (TpFlav, Thaps_19141) and lacks a clade II gene. CRISPR/Cas9 was used to generate three independent KO lines, each with deleted FMN-binding sites (*Figure 3—figure supplement 1A–C*). Two wild type-like lines (WT) were also retained that were transformed with the same plasmid, but in which the flavodoxin gene was not edited (*Figure 3—figure supplement 1B and C*). Cell growth, Fv/Fm, and the final carrying capacity were not significantly different between WT and KO cells under iron-replete growth conditions (*Figure 3A and B*, *Figure 3—figure supplement 1D and E*). Cell counts were similar between WT and KO lines, reaching about 3 million cells per ml at day 12 (WT 3.04±0.25, KO 2.93±0.11, *Figure 3A*, *Figure 3—figure supplement 1D*). Photosynthetic efficiency was also similar between WT and KO cells (0.64±0.016, 0.63±0.040, respectively, at day 1, *Figure 3B*, *Figure 3—figure supplement 1E*). The WT and KO lines displayed comparable reductions in cell division and Fv/Fm under iron limitation and after ~3 d, reached a maximum of about 400,000 cells per ml (*Figure 3A*, *Figure 3—figure supplement 1F*), and decreased in Fv/Fm from 0.63±0.016 (all cell lines in replete conditions) to 0.57±0.016 (KO lines) or 0.57±0.004 (WT lines) (*Figure 3B*, *Figure 3—figure supplement 1G*). Thus, both wild-type and KO *T. pseudonana* lines respond poorly to iron-limiting conditions, which suggests that wild-type cells do not replace their iron-requiring Fd with flavodoxin during acclimation to low iron. A difference in phenotype between WT and KO cell lines emerged, however, after treatment of replete cells with $H_2O_2$ (*Figure 3C*, *Figure 3—figure supplement 1H and I*). Exposure to 100 µM $H_2O_2$ killed ~73% of WT cells after 24 hr. In contrast, exposure of KO cells to the same dose of $H_2O_2$ resulted in the death of ~100% of the KO cells after 24 hr, comparable to what was seen when WT cells were exposed to twice the amount of $H_2O_2$ (200 µM). The hypersensitivity of KO lines to $H_2O_2$ (*Figure 3C*, *Figure 3—figure supplement 1H and I*) indicates that clade I flavodoxins play a role in the oxidative stress response.

## Environmental flavodoxin transcription in the North Pacific Ocean

To expand our analysis beyond laboratory studies and explore the potential roles of clade I and clade II flavodoxins in natural communities, we interrogated a total of 120 eukaryotic surface-water metatranscriptomes sampled during three research cruises in the North Pacific Ocean: Gradients 1 cruise (April/May 2016) transited south/north along ~158°W, from 21°N to 38°N (10 stations, triplicates); Gradients 2 cruise (May/June 2017) transited south/north along ~158°W, from 24°N to 42.5°N (10 stations, triplicates), and a diel sampling cruise (July/August 2015) was conducted in the North Pacific Subtropical Gyre (NPSG), at ~156.5°W, 24.5°N near station ALOHA (24 times points, duplicates); an on-deck nutrient-addition experiment was conducted on Gradients 2 at 37°N, 158°W (incubations: four conditions, triplicates) (*Figure 4A*).

Clade and taxonomic affiliations at either the species or genus level were assigned to each assembled environmental transcript (contig) based on their phylogenetic placement (*Barbera et al., 2019*) within our stramenopile-expanded photosynthetic flavodoxin reference tree (*Figure 4—figure supplement 1*). Contigs for the two flavodoxin clades were detected in metatranscriptomes from all three expeditions, with a greater number and phylogenetic diversity of clade II flavodoxin contigs, particularly along the Gradients 1 and 2 transects (*Figure 4B*). The majority of environmental clade II

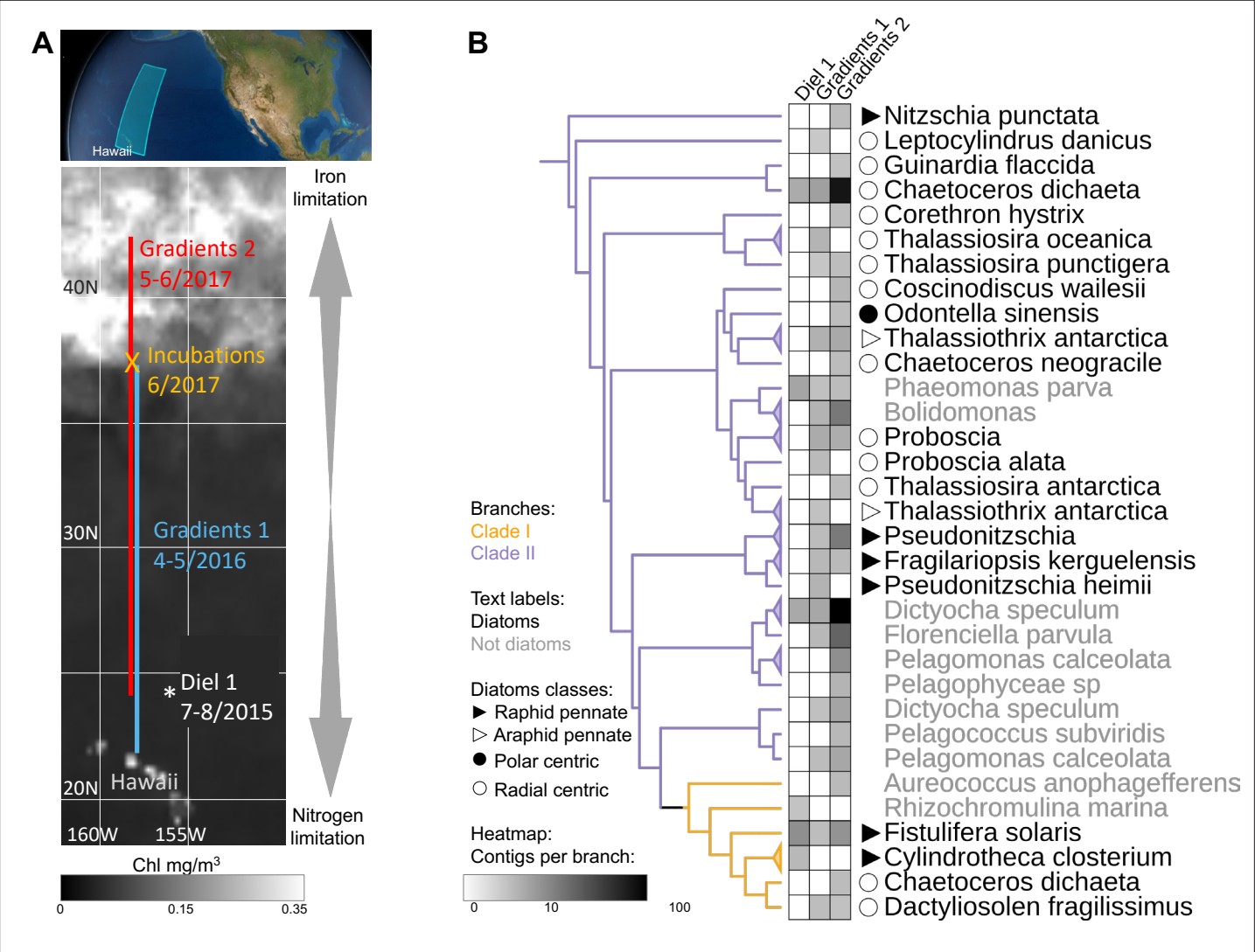

**Figure 4.** Detection of the two flavodoxin clades in the North Pacific. (**A**) Overview of sampling area with a background of satellite, average chlorophyl estimate, from May 9 to June 26, 2017. Data provided by the Ocean Colour Thematic Center at the Copernicus Marine environment monitoring service (CMEMS) and visualized with SimonsCMAP (***Ashkezari et al., 2021***). Cruise dates (month/year) and locations and the site of the on-deck incubation during Gradients 2 are marked. (**B**) Heatmap representing the number of stramenopiles clade I or II flavodoxin contigs detected in each cruise placed on the flavodoxin reference phylogenetic tree. Branches consisting of species (and strains) from the same genus were collapsed and marked with triangles at the edges and labeled according to the highest taxonomic rank. The uncollapsed clade I and clade II region of the reference tree with environmental placements is shown in ***Figure 4—figure supplement 1***. Branch colors represent clade I (orange) and clade II (purple). The genus and species names are in black for diatoms or gray for other stramenopiles.

The online version of this article includes the following figure supplement(s) for figure 4:

**Figure supplement 1.** Flavodoxins in the North Pacific Ocean maximum-likelihood (RAxML) tree of flavodoxins, with placements of environmental reads from the North Pacific Ocean.

flavodoxins were most closely related to the reference sequence of the radial centric diatom *Chaetoceros dichaeta*, originally isolated from the South Atlantic; additional clade II contigs were distributed amongst 11 other diatom genera (***Figure 4B***, ***Figure 4—figure supplement 1***). Within the non-diatom stramenopiles, a majority of clade II flavodoxin contigs were assigned to *Dictyocha* and *Florenciella,* with additional contigs distributed amongst five other genera (***Figure 4B***, ***Figure 4—figure supplement 1***). The clade I flavodoxins were assigned primarily to *Fistulifera solaris*, a small pennate diatom similar to those dominating the diatom population in this area (***Dore et al., 2008***; ***Villareal et al., 2012***); additional clade I contigs were assigned to three other reference diatoms, including *C. dichaeta*, and two non-diatom stramenopiles (***Figure 4B***, ***Figure 4—figure supplement 1***).

Community composition, nutrient availability (*Gradoville et al., 2020*; *Juranek et al., 2020*; *Lambert et al., 2022*; *Park et al., 2022*; *Pinedo-González et al., 2020*), and flavodoxin transcriptional patterns (*Figure 5A and B*, *Figure 5—figure supplement 1*, *Supplementary file 1d*) varied spatially along the Gradients 1 and 2 south/north transects. On Gradients 1, iron (Fe) and nitrate (N) displayed the expected opposing concentration patterns with increasing N and decreasing Fe concentrations north of the NPSG, resulting in low N/Fe (~0.01 µM/nM) within the NPSG and 3–4 orders of magnitude higher N/Fe (~10–100 µM/nM) within the transition zone and further north (*Figure 5C and D*). In contrast, on Gradients 2, Fe increased within the transition zone due to increased dust deposition (*Pinedo-González et al., 2020*), resulting in a relatively low N/Fe (<10 µM/nM) throughout the transition zone up to ~40°N (*Figure 5E and F*). We compared flavodoxin transcript abundances between genera across the two cruise transects (*Figure 5A and B*) by mapping the short sequence reads to our flavodoxin contigs and normalizing these counts to the total transcript concentrations assigned to their respective taxonomic order for each sample (scheme in *Figure 5—figure supplement 1A*). Clade I flavodoxin transcripts were detected for two genera, *Fistulifera* and *Dactyliosolen*, and the relative abundances of these transcripts were higher within the NPSG where N/Fe ratio remained <1 (µM/nM) for both cruises (*Figure 5A–F*). In contrast, the clade II flavodoxin transcript abundances displayed spatial differences between the two cruises. On Gradients 1, relative clade II flavodoxin transcript abundances increased within the transition zone, peaking at 33°N where the N/Fe ratio reached >37 (µM/nM), for all detected genera except *Chaetoceros,* which displayed the highest relative transcript abundances at 37°N, the most northern station of this transect (*Figure 5A, C and D*). Consistent with the higher Fe concentrations detected on Gradients 2 (*Park et al., 2022*; *Pinedo-González et al., 2020*), relative clade II transcript abundance did not undergo as great an increase within the transition zone as seen during the Gradients I cruise. Instead, transcript abundances for 4 of 10 detected genera did not peak until 40°N, the station at which N/Fe first reached >6 µM/nM (*Figure 5B, E and F*). These results indicate that clade II flavodoxins are expressed by diverse genera specifically under relatively high-nitrate, low-iron conditions, whereas clade I flavodoxin transcription is restricted to a few diatom genera in oligotrophic, non-iron-limiting conditions.

To determine how North Pacific diatom genera at a given location respond to changes in N/Fe, we conducted on-deck incubation experiments on Gradients 2 using trace-metal clean seawater collected from 37°N, 158°W where the in situ N/Fe ratio was 0.24 (µM/nM, *Figure 5F*). Seawater samples were maintained in on-deck, temperature-controlled incubators for 4 d with triplicate bottles amended with either no nutrient additions (control); 1 nM FeCl$_3$ (+Fe), to alleviate potential iron limitation; or 5 µM NO$_3$ and 0.5 µM PO$_4$ (+N +P), which was expected to enhance iron limitation. Samples for metatranscriptome analysis were taken at T = 0 and T = 4 d. For this study, the flavodoxin transcripts per liter were summed for each genus in each treatment and normalized to the total flavodoxin transcripts per genus detected in all treatments (e.g. 'row-normalized') (*Figure 5G*, *Figure 5—figure supplement 1A*). Flavodoxins from both clades were detected in these experiments. Clade I flavodoxins transcripts were low in both in situ samples (T = 0) and in the experimental treatments (T = 4 d) (37°N *Figure 5B*, *Supplementary file 1d*) with both nutrient amendment conditions (Fe or N+P) resulting in a reduction in relative transcript abundances assigned to *Chaetoceros* and *Dactyliosolen* (*Figure 5G*). In contrast, relative clade II transcript abundances increased significantly in the N+P amendment treatment compared to both the controls and the Fe amendment in all genera except *Chaetoceros,* *Proboscia,* and *Pseudo-nitzschia* (*Figure 5G*). *Proboscia* displayed the highest relative transcript abundances in situ (T = 0) and *Chaetoceros* and *Pseudo-nitzschia* displayed the highest relative transcript abundances in the control. Together, these results indicate that diatoms can respond to iron limitation by upregulating transcription of clade II flavodoxins, which supports the inference that the relatively low abundance of clade II transcripts detected in situ at 37°N reflected iron-replete conditions at the time of sampling. Importantly, these experiments also illustrate the underlying diversity within diatom genera in their response to iron availability and highlight additional diatoms such as *Pseudo-nitzschia* and *Chaetoceros* for future laboratory studies of adaptive responses to iron-limiting conditions.

To further explore a potential role for clade I flavodoxins, we examined transcriptional patterns of clade I and clade II flavodoxins in metatranscriptomes collected in duplicate every 4 hr for 4 d within the NPSG (Diel 1 cruise, *Wilson et al., 2017*), a low-nitrogen, high-light environment. Each day the photosynthetically active radiation (PAR) reached ~2000 µmol photons·m$^{-2}$·s$^{-1}$ (*Coesel et al., 2021*), conditions expected to enhance oxidative stress. Flavodoxin transcripts per liter were normalized

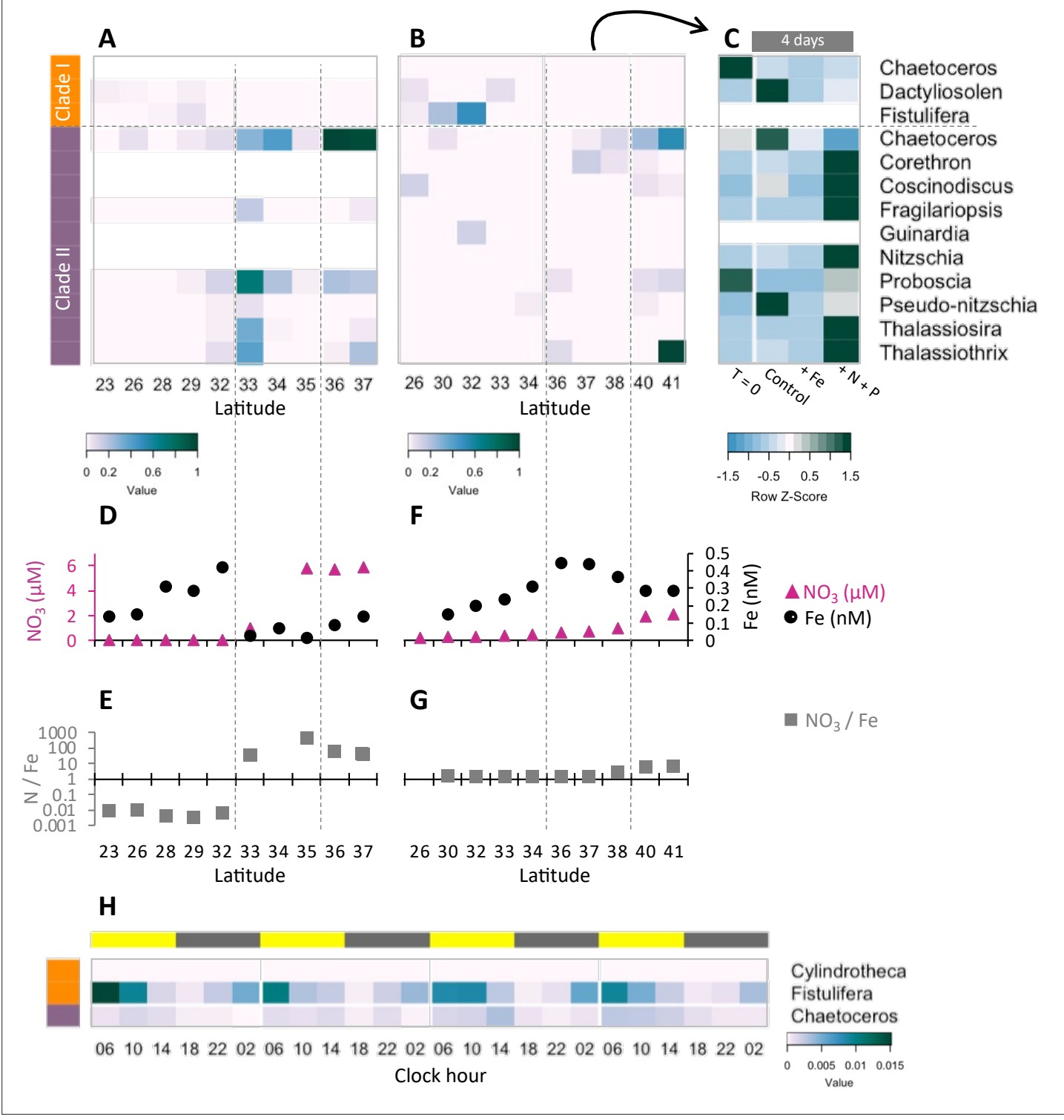

**Figure 5.** Transcriptional patterns of diatom flavodoxin genes in the North Pacific. (**A, B**) Heatmaps of relative diatom flavodoxins transcription across the Gradients 1 transect (**A**) or Gradients 2 transect (**B**). Flavodoxin sequence reads per liter were summed at the genus level and normalized to the cumulative number of reads per liter. White rows indicate where flavodoxin transcripts were not detected. (**C, E**) Total dissolved iron (Fe, black circles) and nitrate ($NO_3$, pink triangles) concentrations across Gradients 1 (**C**) and Gradients 2 (**E**) transects (*Gradoville et al., 2020*; *Juranek et al., 2020*; *Park et al., 2022*; *Pinedo-González et al., 2020*). (**D, F**) Nitrate to iron ratio (gray rectangles) of these transects. (**G**) Flavodoxin transcription following nutrient enrichment incubations conducted at Gradients 2, 37°N. Water were sampled at T = 0 and incubated with no added nutrients (Control) or with 1 nM $FeCl_3$ (+Fe), or 5 µM $NO_3$ and 0.5 µM $PO_4$ (+N +P) and sampled for metatranscriptomes after 4 d. Transcripts per liter are row normalized.

*Figure 5 continued on next page*

Figure 5 continued

(**H**) Diatom relative flavodoxin transcript abundance across the diel cycle, sampled during the Diel 1 cruise. Upper bar represents day (yellow) or night (gray). Side bars represent flavodoxin clade I (orange) or clade II (purple). Note color scale differs from those of (**A, B, G**).

The online version of this article includes the following figure supplement(s) for figure 5:

**Figure supplement 1.** Transcriptional patterns of flavodoxin genes in the North Pacific Ocean.

by the total reads assigned to each taxonomic order, as done for the Gradients transects samples (*Figure 5—figure supplement 1A*). Clade II transcript abundances were low throughout the diel cycle (*Figure 5H*), in agreement with the low transcript levels detected at similar latitudes along the Gradients transects (~24°N, *Figure 5A and B*) representative of the non-iron-limiting conditions at these stations. In contrast, clade I flavodoxins exhibited a clear diel pattern in transcript abundance. Most contigs were assigned to *F. solaris*, and 13 out of 19 contigs had a significant peak in transcript abundance at 6 AM (RAIN analysis, BH correlation <0.05) that then decreased gradually throughout the day and reached a minimum at dusk. The increase in clade I transcript abundances during the night beginning at ~10 PM to 2 AM suggests a diel-regulated anticipation of the high light exposure after dawn.

## Discussion

Marine phytoplankton rely on multiple mechanisms to acclimate to the oxygen-rich, iron-poor conditions of contemporary oceans. We focused on the role of flavodoxin, an iron-independent electron shuttle protein that can replace the more efficient iron-dependent protein ferredoxin under iron-limiting conditions, and which is less sensitive to ROS. Both ferredoxin and flavodoxin were presumably present in the cyanobacteria engulfed during the primary endosymbiosis that gave rise to the plastids of the green, red, and glaucophyte algae (*Campbell et al., 2019*). During the subsequent secondary endosymbiosis of a red alga, these proteins were transferred to stramenopile, dinoflagellate, and haptophyte lineages, which together dominate carbon flux in contemporary global oceans. The complex evolutionary history of flavodoxin has resulted in distributions scattered across all domains of life, a pattern hypothesized to reflect both multiple horizontal gene transfers and gene loss in organisms that evolved in iron-rich environments (*Pierella Karlusich et al., 2015*; *Pierella Karlusich and Carrillo, 2017*). Our expanded genetic survey of flavodoxins in marine microorganisms confirmed the expected relatedness of flavodoxins from cyanobacteria and primary endosymbionts (red and green algae), and their divergence from those of secondary endosymbionts. Responsiveness to iron limitation by both marine cyanobacterial and green algal flavodoxins supports the proposition that iron availability is a strong selective pressure on retention of flavodoxin within the genomes of these organisms (*Pierella Karlusich et al., 2015*). The distribution of the clade I and clade II flavodoxins within secondary (and tertiary) endosymbionts points towards additional selective pressures. Clade II flavodoxins from model organisms display responsiveness to iron limitation, exhibiting the expected canonical replacement of ferredoxin by flavodoxin under iron-limiting conditions. Yet, the distribution of clade II flavodoxins across stramenopiles lineages appears unrelated to contemporary iron conditions. Clade I flavodoxins are restricted to stramenopiles, apparently differentiated in the Diatomista after their divergence from the Chrysista (*Figure 1—figure supplement 1E*). Clade I flavodoxins appear to have been retained across diverse diatoms and are often present in multiple copies, regardless of whether the species was originally isolated from the iron-poor open ocean or the iron-rich coastal environment suggesting alternative selective pressures.

Prior to our study, relatively little attention was given to the potential role of flavodoxin as part of an oxidative stress response in marine phytoplankton. However, in *Escherichia coli*, overexpression of endogenous flavodoxin resulted in enhanced resistance to oxidative stress (*Zheng et al., 1999*), and ectopic expression of cyanobacterial flavodoxin in land plants leads to enhanced tolerance to a broad range of stresses, including oxidative stress (*Blanco et al., 2011*; *Ceccoli et al., 2011*; *Mayta et al., 2019*; *Tognetti et al., 2006*; *Zurbriggen et al., 2008*). In a similar way, when expressed at high levels, the known iron-responsive flavodoxins might mitigate additional stresses that may compromise Fd functionality, such as oxidative stress. Our laboratory studies with five diatom species confirmed that clade II flavodoxin expression is an acclimation to iron limitation, presumably resulting in the replacement of Fd as an electron shuttle in photosystem I. In those diatoms with both clade I and clade II

flavodoxins, only clade II is induced under iron limitation. In those diatoms that lack clade II flavodoxin, the clade I flavodoxin is also induced by iron limitation, although at two orders of magnitude lower transcript abundances (*Figure 2B and C*, *Supplementary file 1a*). This suggests that in those diatoms with only clade I flavodoxins, Fd replacement as a means of reducing iron requirements may play a less important role in acclimation to low-iron conditions. Notably, we used the strong iron chelator DFB to enhance iron limitation in a variety of diatoms, as previously described (*Andrew et al., 2019*; *Kranzler et al., 2021*; *Lampe et al., 2018*; *Timmermans et al., 2001*; *Wells, 1999*), while recognizing that undesirable effects of DFB that are not related to iron limitation per se cannot be ruled out. Here, DFB was used in experiments designed to test whether transcription of the two flavodoxin clades differentially responded to iron limitation. The results from *T. oceanica* and *T. pseudonana* agree with the literature, in which DFB was not added. In *T. oceanica*, only the expression of one clade II flavodoxin was induced (*Figure 2B and C*, as in *Lommer et al., 2012*). The minor induction in mRNA of *T. pseudonana* clade I flavodoxin in response to iron limitation was detected in both long- and short-term adaptation to low iron, without added DFB (*Goldman et al., 2019*; *Thamatrakoln et al., 2012*). This flavodoxin seems to have diel regulation, and the observed induction might be specific to the circadian time and the setting of the diel cycle (*Goldman et al., 2019*).

The knock out of clade I flavodoxin indicates that the clade I flavodoxin partially mitigates $H_2O_2$-induced oxidative stress under iron-replete conditions (*Figure 3C*). Further support for the role of clade I flavodoxin during oxidative stress comes from our observation that the clade I flavodoxins are distinguished from the clade II flavodoxins by replacement of cysteine with alanine at residue 103 (*Figure 1—figure supplement 1F*). Cysteines are susceptible to oxidation under oxidative stress (*D'Autréaux and Toledano, 2007*), which would be expected to inactivate the clade II protein. Future studies in which the oxidative stress is driven by other environmental conditions as supra-optimal irradiation, UV radiation, or biotic interactions are needed to further support the role of clade I flavodoxins in oxidative stress.

The observation that the clade I gene is transcribed at orders of magnitude lower levels than the clade II gene, in cultures and in the North Pacific, suggests either that different transcriptional controls may regulate clade I flavodoxin transcript accumulation within a cell or that oxidative stress may deplete a smaller fraction of the Fd pool compared to iron deprivation. Alternatively, low clade I transcript abundances in our bulk measurements may reflect the previously observed heterogeneity in the response of individual cells to $H_2O_2$ exposure (*Mizrachi et al., 2019*), our means of eliciting oxidative stress. Identification of potential selective pressures for retention of one or both flavodoxin clades will benefit from mechanistic studies with the two proteins to determine whether, as expected, they display different efficiencies under conditions that elicit susceptibility to oxidative stress, and whether their protein levels are differently controlled.

Our metatranscriptomes studies of eukaryotic phytoplankton communities identified both general patterns in transcriptional profiles of the clade I and II flavodoxin genes and highlighted specific responses of different diatoms. Clade II flavodoxins in diatoms in the wild are transcribed in environments where nitrate assimilation is decreased due to iron limitation, defined here by elevated N/Fe (>4 µM N/nM Fe). The different detected diatom genera displayed specific responses to conditions encountered along the two transects. The environmental *Chaetoceros* transcribed clade II flavodoxin exclusively at putative iron-limiting stations, despite detection of this genus at other non-iron-limiting stations. In contrast, two other environmental diatom genera, *Pseudo-nitzschia* and *Fragilariopsis*, were both numerically abundant based on their total transcript abundances (*Figure 5—figure supplement 1C*), and yet the clade II transcript abundance for these two species remained relatively low throughout both transects, suggesting that these species were not experiencing iron limitation at the time of sampling. Similarly, the pelagophyte *Pelagomonas* was also abundant across the transects and yet also transcribed clade II at relatively low levels (*Figure 5—figure supplement 1B and C*). Importantly, several of the detected *Chaetoceros* species encode clade I flavodoxins (*Figure 4—figure supplement 1*), which were either detected at relatively low levels or not detected at all along the transects, indicating the specificity of transcription of the two flavodoxin clades in response to environmental conditions.

The NPSG is an iron-replete, high-light environment, condition that enhances oxidative stress and may thus compromise Fd functionality. Consistent with this premise, the greatest relative abundance of clade I transcripts was detected within the gyre, with a majority of the transcripts assigned to *F.*

*solaris,* which displayed a diel peak in transcript abundance each dawn (**Figure 5H**). This rhythmic pattern was detected only with the clade I flavodoxin from *Fistulifera,* suggesting either a genus-specific adaptation to high light or other oxidative stress generating processes, such as biotic interactions. Interestingly, genome-wide transcriptomic profiles of two model diatoms over the diel cycle also suggest a diel rhythm to clade I flavodoxins transcript abundance. *Phaeodactylum tricornutum* clade I flavodoxin (Phatr3_J13706) expression peaked just before and after dusk (**Smith et al., 2016**), and the benthic diatom *Seminavis robusta* induced the clade I flavodoxins (sro1985_g309400, sro668_g166050) before dusk (**Bilcke et al., 2021**; **Supplementary file 1a**). The diel rhythmicity of clade I flavodoxins in both laboratory studies with model diatoms, as well as the diel rhythmicity in natural diatom communities, emphasizes the potential circadian regulation of this protein in anticipation of the dawn and photosynthetic-related oxidative stress.

Here we show that not all flavodoxins are responsive to low-iron concentrations, and that a subset of flavodoxins (clade I flavodoxins) may be important in oxidative stress response. This specialization might enable diatoms and other stramenopiles to survive in stressful rapidly changing environments. The genetic duplication and functional differentiation of flavodoxin described here add a molecular mechanism that may facilitate diatom survival and adaptation to two major stresses – iron limitation and oxidative stress.

## Materials and methods
### HMM search and phylogenetic tree construction

A custom-made flavodoxin hidden Markov model (hmm)-profile was generated (hmm-build; **Eddy, 1998**) from an alignment containing the Flavodoxin_1 seed alignment (Pfam id PF00258; pfam.xfam.org; **Finn et al., 2016**) amended with the flavodoxin amino acid sequences of *P. tricornutum*, *T. pseudonana,* and *T. oceanica*, using Multiple Alignment using Fast Fourier Transform (MAFFT) version 7.313 (**Katoh et al., 2002**; parameters: `--localpair --maxiterate` 100 –reorder; **Supplementary file 2**; supplementary fasta file 1). Sequences from publicly available genomes and transcriptomes of over 500 marine protists, bacteria, archaea, and viruses (**Coesel et al., 2021**) were searched using the custom hmm-profile (e < 0.001; hmmsearch; **Eddy, 1998**), and hits were clustered at 99% identity by usearch (**Edgar, 2010**). For phylogenetic analysis, sequences were aligned with MAFFT (parameters: `--maxiterate 100 --reorder --leavegappyregion`). A maximum-likelihood phylogenetic tree was built using RAxML version 8.2.4 (**Stamatakis, 2014**; parameters: -f a -m PROTGAMMAWAG -p 451325 -x 12345 -# 100; **Figure 1—figure supplement 1A and B**, **Figure 1A**). As previously noted (**Caputi et al., 2019**), the Pfam and HMM-search do not discriminate those sequences involved in photosynthetic metabolism from other homologous sequences. The phylogenetic tree was used to distinguish clades containing known photosynthetic flavodoxins from the outgroup clade containing non-photosynthetic flavodoxin sequences (labeled in black and red, respectively, **Figure 1—figure supplement 1A**).

A stramenopile-focused phylogenetic analysis was generated by a similar search of 56 additional stramenopile sequences obtained from diverse data sources (**Supplementary file 1b**). The sequences were identified using our custom flavodoxin hmm profile (e < 0.001; hmmsearch) and added to the original alignment using MAFFT (parameters: `--add --localpair --maxiterate 100 --reorder --leavegappyregion`). A fast tree was generated by RAxML (parameters: -f E -p 271321 -m PROTGAMMAWAG) and used to prune branches from the 'outgroup' area (left of the dashed line in **Figure 1—figure supplement 1A**), removing most of the similar clades, and leaving some representatives to attract the environmental reads in the RAxML EPA analysis, described below. The sequences from *Tiarina fusus* were also removed from the tree as this ciliate was reported to be heavily contaminated with prey (**Lasek-Nesselquist and Johnson, 2019**). Additional rogue sequences were detected by RogueNaRok by majority-rule consensus (MR; **Aberer et al., 2013**), and a 'rawImprovement' cut-off of >0.85 was applied to sequences located within the four photosynthetic flavodoxin clades, whereas a more strict cutoff (>0.2) was applied to the outgroup. The remaining 611 sequences were used to generate the stramenopile photosynthetic flavodoxin-focused tree (parameters: -f a -x 25114 -p 269321 -# 100 -m PROTGAMMAWAG) (**Figure 1—figure supplement 1C**, **Figure 1B**). All tree visualizations were performed in the Interactive Tree of Life version 5 (https://itol.embl.de/; **Letunic and Bork, 2021**).

## Culture growth conditions

All diatom isolates were obtained from the Provasoli-Guillard National Center for Culture of Marine Phytoplankton (NCMA). Cultures were grown in 16:8 hr light:dark cycles and a light intensity of about 100 µmol photons·m$^{-2}$·s$^{-1}$. *T. pseudonana* (CCMP1335), *T. oceanica* (CCMP1005), and *A. coffeaeformis* (CCMP1405) were grown at 20°C. *Chaetoceros* sp. (CCMP199), and *C. closterium* (CCMP340) were grown at 24°C. All cultures were adapted to artificial sea water (either EASW or Aquil media) for at least a month before the experiments started. See *Supplementary file 1c* for each species media type and exact growth conditions. The following media were used: f/2 (*Guillard and Ryther, 1962*), L1 (*Guillard and Hargraves, 1993*), EASW (*Berges et al., 2001*), and Aquil (*Price et al., 1989*). For iron limitation experiments, exponentially growing cells were centrifuged at 4000 rpm (5–10 min, 20–22°C as specified in *Supplementary file 1c*). Cells were washed three times with growth media lacking added Fe, and then diluted into either replete media, or media with no added Fe and 1.5 µM desferrioxamine B (DFB; iron chelator, Sigma). Cells were maintained in exponential state by dilutions into fresh media. The cultures were grown in polystyrene flasks to minimize iron contamination. For oxidative stress experiments, we determined the lethal dose for each species in preliminary small-scale experiments and used the lowest lethal concentration (250 µM for *Amphora* and 200 µM for the other cultures) for the transcriptomes. In contrast to the experiments used for transcriptome analysis, all experiments with *T. pseudonana* WT versus the KO lines were done in filtered sea water (FSW) from Puget Sound, supplemented with f/2. Iron limitation was achieved by diluting exponentially growing cells into FSW supplemented with f/2 or f/2 without added iron. Replete cells were also treated with additional concentrations of 25–250 µM of $H_2O_2$.

## Cell counts and cell death

Samples of 150 µl from triplicate flasks were taken and measured immediately using Guava easyCyte 11HT Benchtop Flow Cytometer, excitation by 488 nm laser. Cells were detected by chlorophyl auto-fluorescence (680 ± 30 nm) and forward scatter. The *Amphora* cells formed clumps of cells, which precluded accurate cell counts with the flow cytometer. However, the Fv/Fm measurements indicate that the iron-replete cells were healthy at the time of sampling. Cell death was determined by positive Sytox Green (Invitrogen) staining used at a final concentration of 1 µM. Samples were incubated in the dark for 30 min prior to measurement. Positive gating (525±30 nm) was based according to untreated stained cells and unstained cells.

## Photosynthetic efficiency

Photochemical yield of photosystem II (Fv/Fm) was determined with a Phyto-PAM fluorometer (Heinz Walz GmbH, Effeltrich, Germany) using 15 min dark-adapted cells. Triplicate samples were measured, 1 ml in 1 cm cuvette, each sample was measured three times with 30 s intervals.

## RNA extraction and transcriptome analysis

Cells were harvested by filtration onto 0.22 µm filters. Full details of the number of cells harvested per treatments, per species, and samples that failed library preparation are indicated in *Supplementary file 1c*. The nine samples of each diatom species were sampled together at the same date and time. Filters were snap-frozen in liquid nitrogen and stored at 80°C until extraction. Total RNA was extracted from 0.22 µm membrane filters with the Zymo Direct-zol RNA MiniPrep Plus kit. RNA was quantified using Qbit, and 1900 ng per sample were sent to the Northwest Genomics Center (University of Washington). Poly-A-selection, library preparation, and sequencing were performed at the Northwest Genomics Center with Illumina NextSeq and standard protocols. Sequence reads were trimmed using Trimmomatic 0.39 (*Bolger et al., 2014*) run in paired-end mode with cut adaptor and other Illumina-specific sequences (ILLUMINACLIP) set to TruSeq3-PE.fa:2:30:10:1:true, Leading and Trailing thresholds of 25, a sliding window trimming approach (SLIDINGWINDOW) of 4:15, an average quality level (AVGQUAL) of 20, and a minimum length (MINLEN) of 60. Reads were mapped to the genome of *T. pseudonana* or *T. oceanica* using Hisat2-2.1.0. We calculated the number of *T. pseudonana* and *T. oceanica* reads that aligned to the gene models from resulting sequence alignment map (SAM) files with aligned sequences used in subsequent analyses and normalized using transcripts per million (TPM). De novo transcriptome assemblies were generated from RNA sequences extracted from isolate cultures of *Amphora*, *Chaetoceros*, and *Cylindrotheca*. Each species' quality-controlled

RNA sequencing data was assembled using Trinity v2.12.0 using default settings and with sequences provided as unmerged paired end reads. The pre-assembly RNA sequencing reads were then mapped back to their respective assemblies as a quality control step, and at least 85% of the reads from all cultures successfully mapped onto the resulting assemblies. The assembly resulted in 53,401 contigs for *Amphora*, 37,507 contigs for *Chaetoceros*, and 73,553 contigs for *Cylindrotheca*.

The edgeR pipeline (*Chen et al., 2010*) was used to identify the transcriptional changes in different conditions for all five species and visualized in MDS plot (*Figure 2—figure supplement 1D*). EdgeR was used to detect differential expression of log2 transformed transcript levels based on pairwise comparisons between the control samples and the treated samples. A generalized linear model (GLM) quasi-likelihood F-test (QLTF) was used to test for significant differential expression (p < 0.01 and false FDR < 0.05).

Flavodoxins genes of *T. pseudonana* and *T. oceanica* were taken from the genomes (gene ids: THAPSDRAFT_28635, THAOC_31152, THAOC_19008, THAOC_16623). For *Amphora*, *Chaetoceros,* and *Cylindrotheca,* the flavodoxins sequences of the same species, coastal isolates were used in blast against each species assembly (tblastn, -evalue 0.001). The queries and detected flavodoxins are presented in *Supplementary file 2*; supplementary fasta file 5. These flavodoxins are present in the flavodoxins phylogenetic tree (*Figure 1B*, indicated with red stars, *Supplementary file 1b*, indicated by relevant expression – this study).

## TpFlav gene

The gene sequence and amino acid sequence of *T. pseudonana* flavodoxin were obtained from the JGI genome portal (THAPSDRAFT_28635, transcript ID: EED91575).

## gRNA design for flavodoxin knockout

We combined Golden Gate cloning using two sgRNAs (*Hopes et al., 2017*; *Hopes et al., 2016*), with bacterial conjugation that allows transient transformation and removal of the Cas9 after a deletion is verified (*Karas et al., 2015*). Two sgRNAs were designed to cut ~120 nucleotides, including an FMN-binding site and conserved region (*Figure 3—figure supplement 1A*). Selection of CRISPR/Cas9 targets and estimating on-target score: 20 bp targets with an NGG PAM were identified and scored for on-target efficiency using the Broad Institute sgRNA design program (https://www.broadinstitute.org/rnai/public/analysis-tools/sgrna-design), which utilizes the (*Doench et al., 2014*) on-target scoring algorithm. The sgRNAs that were chosen had no predicted off-targets: the full 20 nt target sequences and their 3′ 12 nt seed sequences were subjected to a nucleotide BLAST search against the *T. pseudonana* genome. Resulting homologous sequences were checked for the presence of an adjacent NGG PAM sequence at the 3′ end. The 8 nt sequence outside of the seed sequence was manually checked for complementarity to the target sequence. In order for a site to be considered a potential off-target, the seed sequence had to match, a PAM had to be present at the 3′ end of the sequence, and a maximum of three mismatches between the target and sequences from the BLAST search were allowed outside of the seed sequence.

## Plasmid construction using Golden Gate cloning

Golden Gate cloning was carried out as previously described (*Weber et al., 2011*) using a design similar to *Graff van Creveld et al., 2021*; *Hopes et al., 2016*. The level 1 (L1) plasmid that enable conjugation was a kind gift from Irina Grouneva, Mackinder lab, University of York, UK (*Nam et al., 2022*). Golden Gate reactions for L1 and level 2 (L2) assembly were carried out using 40 fmol of each component was included in a 20 µl reaction with 10 units of BsaI or BpiI and 10 units of T4 DNA ligase in ligation buffer. The reaction was incubated at 37°C for 5 hr, 50°C for 5 min, and 80°C for 10 min. 5 µl of the reaction was transformed into 50 µl of NEB 5α chemically competent *E. coli*. The nourseothricin resistance gene (NAT) was PCR-amplified from pICH47732:FCP:NAT (Addgene #85984) using primers 1 and 2 (*Supplementary file 1e*) and cloned into a pCR8/GW/ TOPO vector (Thermo Fisher). The FCP promoter, NAT, and FCP terminator level 0 (L0) modules (*Hopes et al., 2016*) were assembled into L1 pICH47751. The sgRNA scaffold was amplified from pICH86966_AtU6p_sgRNA_NbPDS (*Nekrasov et al., 2013*) with sgRNA sequences integrated through the forward primers (3–5, *Supplementary file 1e*). Together with the L0 U6 promoter (pCR8/GW:U6, Addgene #85981), the two sgRNAs were assembled into L1 destination vectors pICH47761 and

pICH47772. Insertion areas in the L1 plasmids were sequenced (6–7, *Supplementary file 1e*). L2 assembly: L1 modules and annealed oligonucleotides (8–9, *Supplementary file 1e*) assembled into the L2 destination vector pAGM4723-Del (Addgene #112207). The final plasmids, L2_Conj_Cas_NAT_Flav, were screened by colony PCR, and the sgRNAs area was sequenced (10–12, *Supplementary file 1e*).

## *T. pseudonana* transformation by bacterial conjugation

The L2_Conj_Cas_NAT_Flav plasmid was delivered to *T. pseudonana* cells via conjugation from *E. coli* TOP10 cells as previously described (*Karas et al., 2015*). The mobilization helper plasmid pTA-Mob, containing all genes necessary for the conjugative transfer of oriT-containing plasmids, was a gift from Rahmi Lale (*Strand et al., 2014*), and was transformed into the cells. *E. coli* pTA-Mob electroporation-competent cells (50 µl) were transformed with 100 ng of the L2_Conj_Cas_NAT_Flav plasmid and used for conjugative delivery of the CRISPR/Cas9 plasmid to the diatom cells. Overnight *E. coli* cultures were inoculated into 50 ml LB +kanamycin (50 mg l$^{-1}$) and gentamicin (20 mg l$^{-1}$), and grown with shaking at 37°C to an OD$_{600}$ of about 0.3. About 40 ml were harvested by 10 min centrifugation at 4000 rpm, 10°C, and resuspended in 400 µl SOC. *T. pseudonana* cells at about 9·10$^5$ cells/ml and Fv/Fm of 0.57 were harvested by centrifugation (4000 rpm, 10 min, 18°C). About 8·10$^7$ cells were suspended in 500 µl of FSW. *T. pseudonana* and *E. coli* cells were mixed, and plated on two plates of 50% FSW L1, 0.8% (w/v), 5% LB (v/v). After drying, the plates were incubated for 90 min at 30°C in the dark, then moved to the regular growth chamber (20°C, about 100 µmol photons·m$^{-2}$·s$^{-1}$) overnight. Then 1 ml of *T. pseudonana* medium was added to each plate and cells were scraped. Cells from each plate were plated on two plates of 50% FSW L1, 0.8% (w/v), nourseothricin 50 or 100 mg·ml$^{-1}$, 0.8% agar plate (selection plates), and incubated at 18°C. Colonies appeared after about 2 weeks.

## Selection of knockout lines

Colonies were scanned for the size of flavodoxin amplicon (primers 13–14, *Supplementary file 1e*), colonies exhibiting double-bands, representing both WT (676 bp) and edited (~530 bp) flavodoxin (probably heterozygotes or mosaic colonies), were re-streaked onto fresh solid medium containing 100 µg·ml$^{-1}$ nourseothricin. Daughter colonies were scanned for the size of flavodoxin amplicon, colonies exhibiting a single band representing biallelic edited flavodoxin were selected for further use. Several colonies exhibiting a single band, either indicative of WT or edited flavodoxins, were transferred into nonselective media for a few weeks. Cells were spread onto nonselective solid media, and single colonies were picked and tested for antibiotic resistance. The flavodoxin gene of nonresistant colonies was sequenced to locate the exact deletion for KO or verify no DNA editing as a control (primers 14–15 *Supplementary file 1e*). WT and two colonies without deletion in the flavodoxin (colonies 5 and 16) are referred to here as 'WT.' Three colonies with deletion in the flavodoxin gene (9, 14, 1) are named here 'KO'.

## Specific cruises sample collection

### Diel 1

Samples were collected from July 25 to August 5, 2015, aboard the R/V *Kilo Moana*, cruise KM1513 from ~156.5°W to 24.5°N. About 7 l seawater samples were pre-filtered through 100 µm nylon mesh and collected onto 142 mm 0.2 µm polycarbonate filters using a peristaltic pump. The cruise was previously described (*Wilson et al., 2017*); additional cruise information and associated data can be found online: https://simonscmap.com/catalog/cruises/KM1513.

### Gradients 1

Samples were collected from April 19 to May 3, 2016, aboard the R/V *Ka'imikai-O-Kanaloa*, cruise KOK1606, from ~23.5 to 37°N, 158°W (*Juranek et al., 2020*). About 6–10 l seawater samples were pre-filtered through a 200 µm nylon mesh and collected by sequential filtration through a 3 µm and a 0.2 µm polycarbonate filters using a peristaltic pump. In this study, we combined the two size fraction in the analysis. Additional cruise information and associated data can be found online: https://simonscmap.com/catalog/cruises/KOK1606.

## Gradients 2

Samples were collected from May 27 to June 13, 2017, aboard the R/V *Marcus G. Langseth*, cruise MGL1704, from ~26 to 41°N, 158°W (*Juranek et al., 2020*). About 6–10 l seawater samples were pre-filtered through a 100 μm nylon mesh and collected as above. Additional cruise information and associated data can be found online: https://simonscmap.com/catalog/cruises/MGL1704.

## Gradients 2 on-deck incubations

At 37°N, 158°W, a total of 20 l seawater was collected into replicate polycarbonate carboys from 15 m depth and incubated at in situ temperature for 96 hr in on-deck, temperature-controlled incubators screened with 1/8 inch light blue acrylic panels to approximate in situ light levels at 15 m. Triplicate carboys were sampled immediately (T = 0) or amended with 1 nM $FeCl_3$ (+Fe), 5 μM $NO_3$, and 0.5 μM $PO_4$ (+N +P). Triplicate carboys with no amendment served as a control. After 4 d, the samples were filtered in the same way as the transect samples (*Lambert et al., 2022*).

All metatranscriptomic samples were flash frozen in liquid nitrogen and subsequently stored at 80°C until further processing. RNA extraction and sequencing was previously described (*Durham et al., 2019*; *Lambert et al., 2022*). Briefly, RNA was extracted with a set of 14 internal mRNA standards in the extraction buffer that were used to correct for extraction and sequencing efficiency and allow final normalization to reads per liter (*Satinsky et al., 2013*). After DNase treatment, purification, and quantification, the eukaryotic mRNAs were poly(A)-selected, sheared, and used to construct complementary DNA libraries. Sequences were de novo assembled and functionally and taxonomically annotated. The metatranscriptome data from the Diel 1 are available through the NCBI's SRA under BioProject PRJNA492142, Gradients 1 under BioProject PRJNA690573, and Gradients 2 under BioProject PRJNA690575.

## Environmental metadata sourcing and processing

Dissolved iron and nitrogen concentrations were measured during the Gradients cruises (*Gradoville et al., 2020*; *Juranek et al., 2020*; *Park et al., 2022*; *Pinedo-González et al., 2020*) and retrieved from the Simons Collaborative Marine Atlas Project pycmap API (CMAP; https://simonscmap.com/).

## Flavodoxins in the metatranscriptomes

The flavodoxin detection and quantification bioinformatic workflow is shown in *Figure 5—figure supplement 1A*. For the three cruises, environmental metatranscriptome contigs with homology to flavodoxin were recruited to the custom-made hmm-profile described above (e < 0.001; hmmsearch). The environmental sequences were aligned to the flavodoxin alignment (stramenopile-enhanced, *Supplementary file 2*; supplementary fasta file 3) using MAFFT (parameters: `--add --local-pair --maxiterate 100 --reorder`) and placed on the flavodoxin reference phylogenetic tree (*Figure 1B*, *Figure 1—figure supplement 1C*) with the additional stramenopiles by RAxML Evolutionary Placement Algorithm (EPA; *Barbera et al., 2019*; parameters: -f v -m PROTGAMMAWAG). Environmental reads with like_weight_ratio >0.8 that were placed on clade I or II stramenopiles at genus level or more specific were kept (*Figure 4—figure supplement 1*). The clade and genus were determined from the flavodoxin reference phylogenetic tree. Reads per liter from each sample were summed to the clade and genus level, and divided by all the reads of the taxonomic order of the contigs, as determined by Lowest Common Ancestor (LCA) using the LCA algorithm in DIAMOND in conjunction with NCBI taxonomy (*Buchfink et al., 2015*), as was previously described (*Groussman et al., 2021*). The normalized counts were averaged for the replicates, and the two size-fractions were summed. Short reads from the incubation experiments were mapped against the flavodoxin contigs from Gradients 2 using kallisto (*Bray et al., 2016*) and normalized to reads per liter using synthetic standards, as in *Coesel et al., 2021*; *Durham et al., 2019*. The normalized reads were aggregated and summed to genus level, the replicates were averaged and the two size-fractions were summed. In *Figure 5G*, the reads are normalized per clade and genus. The difference between this method and the underway stations is shown in *Figure 5—figure supplement 1A*.

## Determining significant diel periodicity

Significant periodicity of the Diel 1 contigs was taken from *Groussman et al., 2021*, with the Rhythmicity Analysis Incorporating Non-parametric Methods (RAIN) package implemented in R (*Thaben*

*and Westermark, 2014*). The p-values from RAIN analyses were ranked and corrected at an FDR < 0.05 using the Benjamini–Hochberg FDR method.

## Acknowledgements

We thank the crew and scientific party of the R/V *Kilo Moana*, R/V *Kaimikai O Kanaloa,* and R/V *Langseth*, and the operational staff of the Simons Collaboration on Ocean Processes and Ecology (SCOPE) program for logistical support. We thank Bryn Durham for assisting with on-deck incubation, Aidan DeHan for assistance in physiological measurements, and Zinka Bartolek for fruitful discussions. This work was supported by a grant from the Simons Foundation (SCOPE Award ID 721244 to EVA).

## Additional information

### Funding

| Funder | Grant reference number | Author |
|---|---|---|
| Simons Foundation | 426570SP | E Virginia Armbrust |
| Simons Foundation | 721244 | E Virginia Armbrust |

The funders had no role in study design, data collection and interpretation, or the decision to submit the work for publication.

### Author contributions

Shiri Graff van Creveld, Conceptualization, Data curation, Formal analysis, Investigation, Visualization, Methodology, Writing - original draft, Writing - review and editing; Sacha N Coesel, Methodology, Writing - original draft, Writing - review and editing; Stephen Blaskowski, Data curation, Software, Writing - review and editing; Ryan D Groussman, Data curation, Software, Investigation, Writing - review and editing; Megan J Schatz, Investigation; E Virginia Armbrust, Conceptualization, Supervision, Funding acquisition, Writing - original draft, Writing - review and editing

### Author ORCIDs

Shiri Graff van Creveld http://orcid.org/0000-0002-3445-3046
Sacha N Coesel http://orcid.org/0000-0001-9422-8388
Stephen Blaskowski http://orcid.org/0000-0003-1621-8955
Ryan D Groussman http://orcid.org/0000-0001-7874-7217
Megan J Schatz http://orcid.org/0000-0003-4669-0740
E Virginia Armbrust http://orcid.org/0000-0001-7865-5101

### Decision letter and Author response

Decision letter https://doi.org/10.7554/eLife.84392.sa1
Author response https://doi.org/10.7554/eLife.84392.sa2

## Additional files

### Supplementary files

• Supplementary file 1. Supplementary tables. (**a**) Diatoms flavodoxins expression from the literature. Species name, original paper, and the specific information about flavodoxin gene ID and the expression in each paper are indicated. (**b**) Information about the species and strains with clade I or clade II flavodoxins. Includes sequence id, taxonomy, data type (genome or transcriptome), isolate location and growth temperature (from culture collections NCMA or RCC when possible, from manuscript describing the isolate when possible). Reference for relevant transcriptome when available, number of flavodoxins from each clade in the phylogenetic trees (# clade I/II). Number of similar flavodoxins that were remove from the alignment due to high sequence similarity (Removed* clade I/II). Strains presented in *Figure 1A* (yes), or within the stramenopiles added for *Figure 1B*. (**c**) Diatom species, with information about the exact isolates, growth, and harvesting conditions for the different transcriptomes done in this study. (**d**) Stramenopiles flavodoxins transcription in the North Pacific, data presented in *Figure 5*. Stramenopiles flavodoxins transcription across the

Gradients 1 transect, R/V *Kaimikai O Kanaloa*, KOK1606 (April/May 2016), and Gradients 2 transect, R/V *Langseth*, MGL1704. Flavodoxin sequence reads per liter were summed at the genus level and normalized to the cumulative number of reads per liter at the order level to compare between genera. Stramenopiles flavodoxin transcription following nutrient enrichment incubations conducted at Gradients 2, station 11, 37°N. Water samples were sample at T = 0 and incubated with no added nutrients (Control) or with 1 nM $FeCl_3$ (+Fe), or 5 µM $NO_3$ and 0.5 µM $PO_4$ (+N + P) sampled for metatranscriptomes after 4 d. Transcripts per liter were summed at the genus level. (**e**). Primers used for Golden Gate cloning, plasmid verification, and deletion scan and sequencing of *T. pseudonana* flavodoxin gene.

• Supplementary file 2. Supplementary fasta files. Supplemental fasta file 1: Amino acid sequence alignments of flavodoxin PF00258 and *P. tricornutum, T. pseudonana,* and *T. oceanica*, used to generate the hmm-profile. Supplemental fasta file 2:Amino acid sequence alignments of flavodoxins used to create the phylogenetic tree in *Figure 1A*. Supplemental fasta file 3: Amino acid sequence alignments of flavodoxins used to create the phylogenetic tree in *Figure 1B*. Supplemental fasta file 4: Amino acid sequence alignments of stramenopiles flavodoxins aligned to *Chondrus crispus* flavodoxin, used for *Figure 1—figure supplement 1F*. All the above are aligned and trimmed. Supplemental fasta file 5: Flavodoxins sequences used to detect flavodoxins in *Amphora, Cylindrotheca,* and *Chaetoceros* transcriptomes, and detected flavodoxins in those species.

• MDAR checklist

## Data availability

Sequencing data from cultures have been deposited in GEO under accession code GSE217467. All other relevant data supporting the findings of the study are available in this article and its Supplementary files.

The following dataset was generated:

| Author(s) | Year | Dataset title | Dataset URL | Database and Identifier |
|---|---|---|---|---|
| Graff van Creveld S, Coesel S, Blaskowski S, Groussman R, Schatz M, Armbrust E | 2022 | Five diatoms transcriptomes under iron limitation and oxidative stress | https://www.ncbi.nlm.nih.gov/geo/query/acc.cgi?acc=GSE217467 | NCBI Gene Expression Omnibus, GSE217467 |

The following previously published datasets were used:

| Author(s) | Year | Dataset title | Dataset URL | Database and Identifier |
|---|---|---|---|---|
| Durham BP, Carlson AK, Groussman LT, Heal RD, Cain KR, Morales KR, Coesel RL, Morris SN, Ingalls RM, Armbrust AE, Virginia E | 2018 | Diel Eukaryotic Metatranscriptomes from the North Pacific Subtropical Gyre | https://www.ncbi.nlm.nih.gov/bioproject/?term=PRJNA492142 | NCBI BioProject, PRJNA492142 |
| Lambert BSG, Schatz RD, Coesel MJ, Durham SN, Alverson BP, White AJ, Armbrust AE, Virginia E | 2021 | Eukaryotic Metatranscriptomes from the North Pacific Ocean (Gradients 1) | https://www.ncbi.nlm.nih.gov/bioproject/PRJNA690573 | NCBI BioProject, PRJNA690573 |
| Lambert BSG, Schatz RD, Coesel MJ, Durham SN, Alverson BP, White AJ, Armbrust AE, Virginia E | 2021 | On-deck Incubation Eukaryotic Metatranscriptomes (Gradients 2) | https://www.ncbi.nlm.nih.gov/bioproject/PRJNA690575 | NCBI BioProject, PRJNA690575 |

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
