## [Editor Report]

This study presents valuable findings, with solid evidence, regarding the functional diversification of flavodoxins from diatoms, a protein initially described as an Fe-sparing substitute for ferredoxin in Fe-poor open ocean environments.

---

## [Decision Letter]

**Decision letter after peer review:**

Thank you for submitting your article "Divergent functions of two clades of flavodoxin in diatoms mitigate oxidative stress and iron limitation" for consideration by *eLife*. Your article has been reviewed by 2 peer reviewers, and the evaluation has been overseen by a Reviewing Editor and Detlef Weigel as the Senior Editor. The following individual involved in the review of your submission has agreed to reveal their identity: Adam Kustka (Reviewer #1).

Essential revisions:

1) A more thorough discussion of some of the nuances and possible limitations of the oxidative stress experiments is warranted.

2) Better discussion of potential unknown effects of inducing iron starvation with a strong chelator, DFB

3) The reviewers highlight several other key areas that must be sufficiently addressed.

*Reviewer #1 (Recommendations for the authors):*

Overall, I thoroughly enjoyed the manuscript.

I had some concerns regarding the transcriptomic experiments. First, the iron limitation is induced by adding a strong Fe chelator, desferrioxamine B, which does not provide consistent growth rate limitation. Expanding on the public comments, in T. pseudonana this resulted in a possible complete cessation of growth (within the resolution of the cell density measurements). Cells that have light-harvesting complexes and are exposed to light but cannot divide may be identified as iron-limited but may experience oxidative stress not related to growth rate limitation by low iron. Taken alone, it is reasonable that this could explain the "low Fe" response of flavodoxin that was not observed in previous studies referenced in Table S1. Also, neither the low iron nor the control treatment seems to have supported growth in Amphora in the days prior to transcriptome sampling. The Chaetoceros experiments seem reasonable.

*Reviewer #2 (Recommendations for the authors):*

General comments:

The authors should mention that overall Clade II is much more highly expressed than clade I flavodoxin in the North Pacific, even if the goal of the study is to identify the function of Clade I flavodoxin. To make this point clear, Figure S4 could be added to the main manuscript, substituting it for another less informative figure. Eg. Figure 4 in the main paper is difficult to interpret because of the lack of context with the iron concentrations. It is difficult to see without the iron concentration data (which is presented in Figure 5c-f), whether the pattern of clade I and clade II flavodoxin expression is followed with regard to their responsiveness to iron limitation.

At the end of the manuscript, one is left with the question of whether or not a KO mutant of the clade II flavodoxin in T. oceanica would also be sensitive to oxidative stress.

Specific comments:

Line 193: a lethal dose of H2O2 was used in the experiment. A dose-response curve would have been more convincing because as is, one cannot judge the tolerance of the KO mutant to oxidative stress in a gradual increase.

---

## [Author Response]

Essential revisions:1) A more thorough discussion of some of the nuances and possible limitations of the oxidative stress experiments is warranted.

We now clarify in the Results section (L185-188) that our use of exogenous H2O2 additions was based on previous studies with *Phaeodactylum* and *T. pseudonana* that indicate that exogenous addition of μM range of H2O2 is representative for other oxidative stress-responses (Graff van Creveld, 2015, Volpert 2018, Mizrachi 2019). We further added a broader range of H2O2 concentrations according to the reviewer suggestion (Figure S3H). We also acknowledge in the Discussion section (L417-419) that future studies with in which the oxidative stress is driven by other environmental conditions as supra-optimal irradiation, UV radiation or biotic interactions are necessary to strengthen the role of clade I flavodoxins in oxidative stress.

2) Better discussion of potential unknown effects of inducing iron starvation with a strong chelator, DFB.

We added the following paragraph to the Discussion section (L395-410):

“Notably, we used the strong iron chelator DFB to enhance iron limitation in a variety of diatoms, as previously described (Andrew et al., 2019; Kranzler et al., 2021; Lampe et al., 2018; Timmermans et al., 2001; Wells, 1999), while recognizing that undesirable effects of DFB, that are not related to iron-limitation *per se* cannot be ruled out. Here, DFB was used in experiments designed to test whether transcription of the two flavodoxin clades differentially responded to iron limitation. The results from *T. oceanica*, and *T. pseudonana* agree with the literature, in which DFB was not added. In *T. oceanica* only the expression of one clade II flavodoxin was induced (Figure 2B-C, as in Lommer et al., 2012). The minor induction in mRNA of *T. pseudonana* clade I flavodoxin in response to iron limitation was detected in both long- and short-term adaptation to low iron, without added DFB (Goldman et al., 2019; Thamatrakoln et al., 2012). This flavodoxin seems to have diel regulation, and the observed induction might be specific to the circadian time and the setting of the diel cycle (Goldman et al., 2019).”

3) The reviewers highlight several other key areas that must be sufficiently addressed.

We addressed the reviewer comments in the following point-to point response below.

Reviewer #1 (Recommendations for the authors):Overall, I thoroughly enjoyed the manuscript.I had some concerns regarding the transcriptomic experiments. First, the iron limitation is induced by adding a strong Fe chelator, desferrioxamine B, which does not provide consistent growth rate limitation. Expanding on the public comments, in T. pseudonana this resulted in a possible complete cessation of growth (within the resolution of the cell density measurements). Cells that have light-harvesting complexes and are exposed to light but cannot divide may be identified as iron-limited but may experience oxidative stress not related to growth rate limitation by low iron. Taken alone, it is reasonable that this could explain the "low Fe" response of flavodoxin that was not observed in previous studies referenced in Table S1.

We appreciate the thoughtful comment. As described in our comments to the public reviews, we added text to the manuscript that DFB was used in the diatom “survey” experiments to determine whether the different diatom isolates transcribed the two flavodoxin clades in response to iron limitation (L177-179). The experiments that compared the responses of *T. pseudonana* WT and KO lines did not rely on use of DFB and instead cells were diluted into media with no added iron. This mild iron limitation, without strong Fe chelator or the use of artificial seawater, also led to growth arrest of *T. pseudonana* after few days (Figure S3F). Additionally, our results agree with other studies including Thamatrakoln et al., 2012, Goldman et al., 2019 (L395-410).

Also, neither the low iron nor the control treatment seems to have supported growth in Amphora in the days prior to transcriptome sampling. The Chaetoceros experiments seem reasonable.

The *Amphora* formed clumps of cells / short chains that were not well suited to our flow-cytometry-based cell counts. We have added this extra information now to the revised manuscript (L531-534):

“The *Amphor*a cells formed clumps of cells, which precluded accurate cell counts with the flow cytometer. However, the Fv/Fm measurements indicate that the iron replete cells were healthy at the time of sampling.”

Reviewer #2 (Recommendations for the authors):General comments:The authors should mention that overall Clade II is much more highly expressed than clade I flavodoxin in the North Pacific, even if the goal of the study is to identify the function of Clade I flavodoxin. To make this point clear, Figure S4 could be added to the main manuscript, substituting it for another less informative figure. Eg. Figure 4 in the main paper is difficult to interpret because of the lack of context with the iron concentrations. It is difficult to see without the iron concentration data (which is presented in Figure 5c-f), whether the pattern of clade I and clade II flavodoxin expression is followed with regard to their responsiveness to iron limitation.

We agree with the reviewer’s assessment about the differences between clade I and II transcript abundances as noted in the results (L268-270):

“Contigs for the two flavodoxin clades were detected in metatranscriptomes from all three expeditions, with a greater number and phylogenetic diversity of clade II flavodoxin contigs, particularly along the Gradients 1 and 2 transects (Figure 4B).”

We now added the difference in clade I and clade II expression levels in the North Pacific in the discussion (L420-422):

“The observation that the clade I gene is transcribed at orders of magnitude lower levels than the clade II gene, in cultures and in the North Pacific, suggests either that different transcriptional controls may regulate clade I flavodoxin…”

We think that Figure 4B should stay the main figure, as it summarizes Figure S4. Figure 4B is simpler, more readable (with larger labels) and more quantitative compared to Figure S4, thus more suitable for the main text. Both Figures show the taxonomy and clades of the detected flavodoxins in the North Pacific, by the three different expeditions. As the reviewer mentioned, the detailed flavodoxin expression, related to iron concentration is presented in Figure 5.

Figure 4A provides context to Figure 4B and Figure 5, by presenting the sampling area and the relative location of the different expeditions, and on-deck-incubation.

At the end of the manuscript, one is left with the question of whether or not a KO mutant of the clade II flavodoxin in T. oceanica would also be sensitive to oxidative stress.

We agree and would love to have this mutant! However, transformation is not (yet) established for *T. oceanica.* Establishing a new transformation system is out of the scope of the current manuscript.

Specific comments:

Line 193: a lethal dose of H2O2 was used in the experiment. A dose-response curve would have been more convincing because as is, one cannot judge the tolerance of the KO mutant to oxidative stress in a gradual increase.

Thank you for this suggestion. We preformed the requested dose-response to H2O2 and added new Figure S3H, that includes WT and KO responses in 0, 25, 50, 75, 100, 150, 200, 250 µM treatments.